# The Diverse Roles of γδ T Cells in Cancer: From Rapid Immunity to Aggressive Lymphoma

**DOI:** 10.3390/cancers13246212

**Published:** 2021-12-09

**Authors:** Susann Schönefeldt, Tamara Wais, Marco Herling, Satu Mustjoki, Vasileios Bekiaris, Richard Moriggl, Heidi A. Neubauer

**Affiliations:** 1Institute of Animal Breeding and Genetics, University of Veterinary Medicine Vienna, 1210 Vienna, Austria; susann.schoenefeldt@vetmeduni.ac.at (S.S.); tamara.wais@meduniwien.ac.at (T.W.); richard.moriggl@vetmeduni.ac.at (R.M.); 2Department of Hematology, Cellular Therapy and Hemostaseology, University of Leipzig, 04103 Leipzig, Germany; marco.herling@medizin.uni-leipzig.de; 3Hematology Research Unit Helsinki, Helsinki University Hospital Comprehensive Cancer Center, 00290 Helsinki, Finland; satu.mustjoki@helsinki.fi; 4iCAN Digital Precision Cancer Medicine Flagship, 00014 Helsinki, Finland; 5Translational Immunology Research Program and Department of Clinical Chemistry and Hematology, University of Helsinki, 00014 Helsinki, Finland; 6Department of Health Technology, Technical University of Denmark, 2800 Kongens Lyngby, Denmark; vasbek@dtu.dk

**Keywords:** cancer immunity, γδ T cells, γδ T-cell lymphoma, immunotherapy, targeted therapy, γδ T-cell transformation

## Abstract

**Simple Summary:**

γδ T cells play important roles in cancer immunity. Their rapid activation and cytotoxic nature make them promising candidates for use in cell-based immunotherapies; however, under certain conditions, they can induce pro-tumour functions. Furthermore, upon transformation, γδ T cells can develop into aggressive lymphomas with a poor prognosis and no curative therapeutic options. Here, we provide a comprehensive summary of our current knowledge on the complex roles of γδ T cells in cancer. We discuss their anti- and pro-tumour functions in both solid and blood cancers, highlighting the key subsets involved and their potential utility in anti-cancer immunotherapy. We also discuss the mechanisms of γδ T-cell transformation, summarising the resulting γδ T-cell leukaemia/lymphoma entities and their genetic and molecular profiles, as well as current and future treatment strategies.

**Abstract:**

γδ T cells are unique players in shaping immune responses, lying at the intersection between innate and adaptive immunity. Unlike conventional αβ T cells, γδ T cells largely populate non-lymphoid peripheral tissues, demonstrating tissue specificity, and they respond to ligands in an MHC-independent manner. γδ T cells display rapid activation and effector functions, with a capacity for cytotoxic anti-tumour responses and production of inflammatory cytokines such as IFN-γ or IL-17. Their rapid cytotoxic nature makes them attractive cells for use in anti-cancer immunotherapies. However, upon transformation, γδ T cells can give rise to highly aggressive lymphomas. These rare malignancies often display poor patient survival, and no curative therapies exist. In this review, we discuss the diverse roles of γδ T cells in immune surveillance and response, with a particular focus on cancer immunity. We summarise the intriguing dichotomy between pro- and anti-tumour functions of γδ T cells in solid and haematological cancers, highlighting the key subsets involved. Finally, we discuss potential drivers of γδ T-cell transformation, summarising the main γδ T-cell lymphoma/leukaemia entities, their clinical features, recent advances in mapping their molecular and genomic landscapes, current treatment strategies and potential future targeting options.

## 1. Introduction

γδ T cells are a unique set of ‘unconventional’ T cells that reside at the interface between innate and adaptive immunity [1]. The primary functions of γδ T cells are to provide rapid responses to ensure tissue integrity, maintain immune and tissue homeostasis [2,3], sense and fight against cancer and uphold critical host defence and barrier function towards foreign antigens [4,5,6]. These roles are facilitated by their unique tissue distribution; whereas conventional αβ T cells mainly reside in lymphoid organs, γδ T cells preferentially home to and reside in mucosal and epithelial tissues of peripheral organs, which are not usually under surveillance by αβ T cells in the absence of a specific activation signal. Rapid γδ T-cell activation occurs via direct detection of ligands by the T-cell receptor (TCR) or other activating receptors in a major histocompatibility complex (MHC)-independent manner, which again sets them apart from αβ T cells [7].

With respect to their TCR assembly, unlike αβ T cells that have randomly-paired receptor chains, the γ and δ chain pairing is restricted to only a few common combinations, resulting in distinct γδ T-cell subsets [8]. The genes for the γ TCR (*TCRG*) are located on chromosome 13 in mice and chromosome 7 in humans, and genes for the δ TCR (*TCRD*) are found on chromosome 14 in both mice and humans [9]. Between mouse and human, there are species-specific variations in the chain rearrangements [10]. As such, the consensus regarding γδ T-cell nomenclature is to define human γδ T cells by their δ chain segment and mouse γδ T cells by their γ chain segment. In mice, the most commonly found γδ T cell subtypes include Vγ1, Vγ4, Vγ5, Vγ6 and Vγ7 (nomenclature by Heilig and Tonegawa) [11]. In humans, the main γδ T-cell subsets can be divided into Vδ1, Vδ2 and the very minor Vδ3-expressing subsets (nomenclature by Lefranc and Rabbitts) [12] (Figure 1). Importantly, specific Vγ and Vδ chain combinations allow for the recognition of different types of ligands, facilitating the distinct γδ T-cell functions needed in different resident tissues [7]. Interestingly, the Vδ2 chain preferentially pairs with the Vγ9 chain, whereas the Vδ1 and Vδ3 chains do not have such Vγ chain preferences. In contrast to the other subtypes, the Vγ9Vδ2 subset is the predominant γδ T-cell subtype in the peripheral blood (making up ≈50–90% of γδ T cells) and is, to date, the most investigated for their role in health and disease, especially in cancer and immunotherapy [13,14,15,16,17].

Functionally, γδ T cells can be rapidly activated by specific receptor–ligand interactions [7], upon which they possess the capacity for clonal expansion and provide a major source of cytokine and chemokine production [18,19,20,21]. This facilitates their abilities to exert direct or indirect immune responses by functioning as professional antigen presenting cells (APCs) [22,23], providing T- and B-cell helper functions [24] or directly killing compromised cells [25,26,27]. Importantly, γδ T cells can rapidly respond to specific ligand types that are not recognised by other immune cells, where stress signal recognition by γδ T cells as one example in particular is critical for the immune detection of many cancers [28,29]. Consequently, γδ T cells provide an important, active link between the adaptive and innate arms of the immune system, making them key effector cells in cancer immunity [8].

In this review, we summarise the complex roles of γδ T cells in cancer, from their clear anti-tumour activities in both solid and haematological cancers, to their intriguing, dichotomous roles in maintaining and promoting tumour growth. Finally, we also discuss the aggressive lymphomas that can arise when γδ T cells themselves become malignant, highlighting the key genetic features of these cancers and summarising the latest treatment directions.

**Figure 1 cancers-13-06212-f001:**
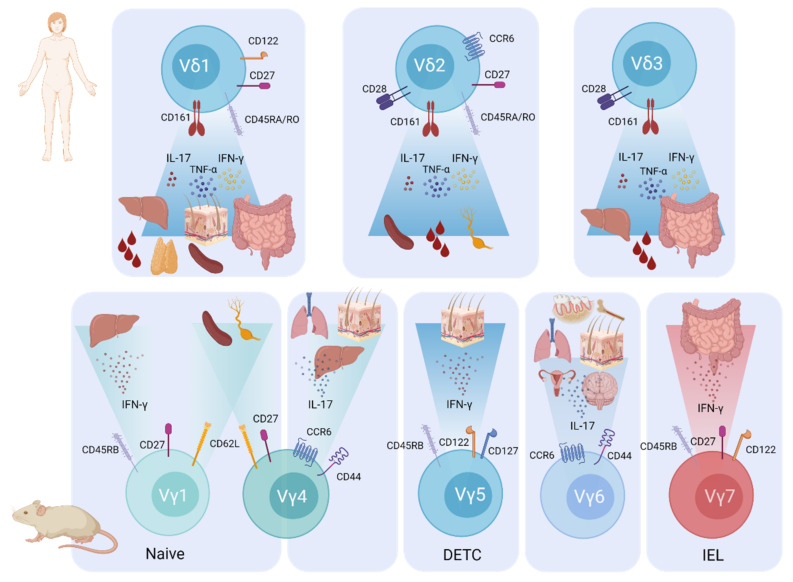
Attributes of γδ T-cell subsets most commonly found in humans or mice. The main human γδ T-cell subsets, Vδ1, Vδ2 and Vδ3 (top), and the main murine γδ T-cell subsets, Vγ1, Vγ4, Vγ5, Vγ6 and Vγ7 (bottom), are shown. Key surface molecules, dominant cytokines and main tissue distributions are depicted [10]. Top: Human γδ T cells can be positive for CD161 (IL-17-producers) [30], CCR6, CD28, CD27, CD122 and CD45RA/RO surface markers, depending on the immunological and tumour microenvironmental context. CD45RA and CD27 surface expression can define specific subtypes: naïve (CD45RA^+^ CD27^+^), memory (CD45RA^−^ CD27^+^), activated effector memory (CD45RA^−^ CD27^−^) and terminally differentiated (CD45RA^+^ CD27^−^) cells [31,32]. Human γδ T cells are predominantly IFN-γ- and TNF-α-producing effectors; however, all subsets can produce IL-17 under certain conditions. Bottom: Mouse Vγ1^+^ T cells are largely IFN-γ-producing and have CD45RB and CD27 surface expression. Vγ4^+^ T cells are predominantly IL-17-producing and express CCR6 and CD44. Both Vγ1^+^ and Vγ4^+^ T cells can also home to secondary lymphoid tissues (expressing CD62L). IFN-γ-producing Vγ5^+^ dendritic epidermal T cells (DETCs) express CD45RB, CD122 and CD127. Vγ6^+^ T cells are primarily IL-17-producing and are positive for CD44 and CCR6. Vγ7^+^ intraepithelial lymphocytes (IELs) expressing CD45RB, CD27 and CD122 primarily secrete IFN-γ. The nomenclature used for T-cell receptor genes is based on the Heilig and Tonegawa system for mouse and the Lefranc and Rabbitts system for human γδ T cells [11,12]. CCR, CC chemokine receptor; CD, cluster of differentiation; IFN-γ, interferon-γ; IL, interleukin; TNF-α, tumour necrosis factor-α.

## 2. Functional γδ T-Cell Subsets in Mice and Humans

The faceted roles of γδ T cells in cancer are shaped by multiple regulators that determine whether they take on a pro- or anti-tumour role, very much depending on their initial effector type programming [33,34,35], the tumour microenvironmental context [36,37,38], encounters with other tumour-associated immune cells [39,40] and the tumour entities themselves [41,42]. Indeed, γδ T cells are critical cells in cancer as well as in general immunity [2,43]; they are the first T cells to develop [44], leave the embryonic thymus and home to their target tissues [45]. Here, they are found throughout the entirety of life as deeply embedded into the cellular organisational structure [46,47].

Generally, murine γδ T cells can be functionally subtyped into RAR-related orphan receptor gamma (RORγt)-programmed interleukin-17A (IL-17)-producing γδ T (γδT17) cells [10,43] and T-box 21 (T-bet)-programmed interferon-γ (IFN-γ)-producing γδ T (γδT1) cells [48]. There is a clear association between TCR Vγ chain usage and effector subtype: murine Vγ1^+^, Vγ5^+^ and Vγ7^+^ γδ T cells are biased towards IFN-γ production, whereas Vγ4^+^ and Vγ6^+^ γδ T cells largely produce IL-17 (Figure 1). Three separate factors have been proposed to account for fate decisions of murine γδT17 or γδT1 effector lineage development in immature thymocytes: (i) TCR signal strength [49], (ii) Notch signalling [50] and (iii) cytokine encounter during the double negative DN3 development stage [45,50]. Notably, while strong γδ TCR signalling can prohibit γδT17 in favour of γδT1 lineage commitment, additional Notch signalling and the addition of certain cytokines can skew the functional phenotype towards the γδT17 lineage [49]. Specifically, Sumaria et al., found in mice that while Notch signalling alone was not able to induce the γδT17 lineage, it was still a requirement, and only in combination with γδT17-associated cytokines (IL-1β, IL-21 and IL-23) was it able to induce the transcriptional network for IL-17 production [49]. As a very recent concept, Lopes et al. found that TCR signal strength also determines the metabolic programming of γδ T cells during thymic development in mice, coinciding with their effector lineage commitment [34]. Specifically, the authors showed that γδT17-programmed cells solely utilise oxidative phosphorylation, whereas strong TCR signalling and initiation of the γδT1 program induces a switch to glycolysis [34]. These subset metabolic requirements could provide interesting new implications for improving γδ T-cell-based immune therapy. We have also recently shown that γδ T-cell effector subtypes are regulated differentially by the two STAT5 proteins, STAT5A/B, using specific gain- or loss-of-function transgenic mouse models [33]. Herein, dominant STAT5A signalling promoted RORγt expression, whereas STAT5B signalling favoured T-bet upregulation, resulting in lineage skewing towards either γδT17 by STAT5A or γδT1 by STAT5B, revealing non-redundant roles of STAT5 homologs in fine-tuning γδ T-cell lineage commitment [33].

In contrast to murine γδ T cells, there is no association between TCR Vδ chain usage and the functional subtype of human γδ T cells; Vδ1, Vδ2 and Vδ3 γδ T cells are present most commonly as the IFN-γ-producing γδT1 subtype [32,51], while no dedicated human γδT17 subset has been identified. There is, however, compelling evidence that under certain circumstances, all human γδ T cells can produce IL-17 [52,53,54,55]. For example, it was shown that a subpopulation of CD28^+^ Vγ9Vδ2 cells produce IL-17, are positive for CD27 and CCR6 and express type-3 signature genes such as *RORC* and *IL23R* [56]. Notably, the recent identification of γδT17 Vδ2 cells in human embryonic thymus also suggests a potential pre-programing mechanism similar to mice [57].

Plasticity, or polarisation, is another important aspect of γδ T-cell physiology [58], resulting in effector function switching even after thymic selection and initial phenotypic programming [33]. Interestingly, polarisation and activation patterns usually occur in an inflammatory or cancer-immunity context [59,60,61]. These processes are often triggered by cytokine stimulation and coupled with TCR stimulation, leading to changes in transcription factor expression and resulting in altered expression of effector molecules and cytokines. Indeed, the right stimulus on a resting (non-activated) γδ T cell can promote switching between the effector subtypes. Aside from the two main subtypes, γδT1 and γδT17, other putative populations have been reported (e.g., γδ T follicular helper (Tfh)-like cells [62] and forkhead box P3 (FOXP3)^+^ γδ T regulatory (Treg)-like cells [63]). In the cancer context, the main extrathymic regulator of γδ T-cell plasticity is the tumour microenvironment (TME) [38]. As discussed further below, the TME provides an abundance of cytokines as well as hypoxic conditions (in the case of many solid cancers) [42,64,65,66,67], contributing to the polarisation of γδ T cells into different effector subtypes and thereby promoting either anti- or pro-tumour functions [64,65,68].

The dichotomy of γδ T cells in cancer immunity lies in the fact that, in general, the γδT1 effector subtype is associated with anti-tumour roles, whereas γδT17 cells have been extensively shown in murine studies to possess pro-tumour functions. In humans, it has been until recently very difficult to study this subtype due to their rarity, tissue-resident nature and lack of protocols for their in vitro expansion. However, recent advancements in this area have facilitated the study of γδT17 cells and their pro-tumour functions in humans [43]. Herein, we focus mainly on the roles of human γδ T-cell subsets in cancer immunity, where possible.

## 3. Anti-Tumour Functions of γδ T Cells

γδ T-cell anti-tumour responses can be categorised into: cellular fitness and stress ligand sensing, death receptor engagement, antibody dependent cytotoxicity (ADCC) and execution of T helper functions (Figure 2). These functions of γδ T cells are facilitated by their rapid cytotoxic activity upon stimulation with specific ligands of their TCR in conjunction with other activating receptors, including Toll-like receptor (TLR) [69], CD16 (FcγRIIIA) [70], CD226 (DNAX accessory molecule-1 [DNAM-1]) [71], CD28 [72], natural killer group 2 member D (NKG2D) [73,74] and natural killer receptors [75] (NKRs; NKp30 [76], NKp44 [77] and NKp46 [73,78]). Activating ligands include non-peptide substances such as phosphoantigens (PAgs) [79], aminobisphosphonates [80] and alkylamines [81,82] as metabolites of the mevalonate pathway upregulated in cancer cells [83], as well as MHC class I polypeptide-related sequence A/B (MICA/MICB) [75,84], heat shock proteins [85], butyrophilins/butyrophilin-like (BTN/BTNL) molecules [20,21] and members of unique long 16 (UL-16)-binding proteins (ULBP/RAET1) [86,87].

Interestingly, cellular stress can be perceived differently by the two main human γδ T-cell subtypes. While specific Vδ1 TCR ligands remain elusive, these cells are highly efficient in detecting cellular stress signals (e.g., stress-induced ligands or glycolipids [85,88] and glycans [89,90]). The activation of Vδ1^+^ cells by TCR engagement combined with cytokine stimulation (i.e., IL-2 or IL-15) upregulates NKRs, correlating with CD56 expression and potent anti-tumour cytotoxicity [76,91]. Both Vδ1 and Vδ2 T cells can detect MICA/B and ULBP ligands on cancer cells via interaction with their NKG2D receptor [92,93], facilitating their activation [74,75] and leading to the release of perforin and granzyme B [93,94]. Interestingly, the crystallisation of the Vδ1 TCR revealed a direct interaction with MICA, although this interaction is of a significantly lower affinity than that of the NKG2D–MICA interaction [84]. On the other hand, it is well established that Vδ2^+^ T cells can respond strongly to PAgs such as isopentenyl pyrophosphate (IPP), which is a metabolite of the mevalonate or microbial deoxyxylulose phosphate pathway [7,95,96]. Cancer cells often have a dysregulated mevalonate pathway, resulting in intracellular accumulation of IPP, which can be sensed by Vγ9Vδ2 T cells, leading to their activation and anti-tumour functions. This mechanism involves IPP binding to the intracellular domain of the butyrophilin molecule BTN3A1 expressed on cancer cells, which interacts with BTN2A1 to bind and activate the TCR on Vγ9Vδ2 cells [97,98]. IPP can also be secreted from cancer cells and APCs into the extracellular environment to activate Vγ9Vδ2 T cells, and this mechanism was shown to involve the ATP-binding cassette transporter A1 (ABCA1) in cooperation with apolipoprotein A-I (apoA-I) and BTN3A1 [18]. Notably, γδ T-cell ligand interactions can vary between mice and humans. For example, PAgs are not recognised by murine γδ T cells due to the lack of an equivalent Vγ9Vδ2 subset [10,29,96,99]. This is important to consider when using models to understand γδ T-cell biology as well as developing γδ T-cell-based cancer immunotherapy. This is currently being addressed by the use of humanised murine models (e.g., that express human receptor molecules on γδ T cells) [21,28,29,100,101].

γδ T cells can also mediate direct cytotoxicity via the engagement of death receptor pathways, acting upon tumour necrosis factor-related apoptosis-inducing ligand (TRAIL) and FAS ligand (FASL) [102,103] (Figure 2). Supporting this, Tawfik et al., demonstrated that reduced TRAIL-R4 expression in cancer cells rendered them less sensitive to Vδ1- and Vδ2-mediated cytotoxicity in an ERK/COX2-dependent manner [27].

Another γδ T-cell mechanism to exert anti-tumour function is through antibody-dependent cellular cytotoxicity (ADCC). Here, a tumour antigen-specific antibody binds to the tumour cell, allowing the Fc segment to be bound by the FcγRIIIA receptor on γδ T cells, inducing γδ T-cell activation and killing of the tumour cell. This mechanism can be exploited to target and kill tumour cells using specific targeting antibodies (e.g., anti-CD20), which were shown to enhance the cytotoxicity of stimulated/expanded Vγ9Vδ2 cells [104]. γδ T cells can also further indirectly promote ADCC by acting as regulators/helpers of the peripheral B-cell repertoire [24], thereby supporting humoral immunity against tumour cells. In a murine model, it was shown that in the absence of CD4^+^ αβ T-cell populations, γδ T cells may support B germinal centre formation [105]. A more recent study reported that γδ T cells could shape B-cell maturation by directly affecting the transitional stages of marginal zone B cells, which was dependent on the Vγ TCR chain expressed [106]. For instance, a loss of Vγ4 and Vγ6 T cells led to a stark loss of peripheral B-cell populations [106]. Interestingly, a subset of human Vγ9Vδ2 T cells isolated from peripheral blood were found to express C-X-C chemokine receptor type 5 (CXCR5) and could, upon stimulation, express the costimulatory molecules ICOS and CD40L, inducing the production of B helper cytokines such as IL-2, IL-4, IL-10 and IL-21 to benefit B-cell functions [107,108]. Similarly, Vδ3^+^ T cells were also shown to upregulate CD40, CD86 and HLA-DR, leading to the production of IgM by B cells [109]. Taken together, these studies demonstrate that γδ T cells can provide B-cell helper function in support of antibody production and class switching, which supports ADCC anti-tumour functions (Figure 2).

γδ T cells can also act as excellent T helper cells to support other immune cell types involved in anti-cancer immunity. For one, they have the capacity to function as professional APCs [110], upregulating the scavenger receptor CD36 to facilitate the uptake of tumour antigens mediated via mitogen-activated protein kinase (MAPK) and NF-κB pathway signalling [111,112]. In turn, this upregulates the co-stimulatory molecules CD40, CD80 and CD86, as well as the MHC class II molecule HLA-DR on γδ T cells to facilitate the activation of cytotoxic CD8^+^ T cells [113,114]. Another aspect of their helper function is the interaction between γδ T cells and dendritic cells (DC). This is a reciprocal relationship, whereby mature DCs can induce the activation and proliferation of γδ T cells [115,116] via the release of IPP [18], and γδ T cells can in turn promote the maturation of DCs [117] via secretion of IFN-γ and TNF-α to increase DC expression of CD86 and MHC-I molecules [118,119] (Figure 2). It was also found that Vδ3^+^ T cells can be stimulated to release certain Th1 (IFN-γ, TNF-α), Th2 (IL-4) and Th17 (IL-17) cytokines to induce DC maturation [120].

### 3.1. γδ T Cells in Solid Cancers

In human solid cancers, γδ tumour-infiltrating lymphocytes (TILs) have been proposed by many studies to be relevant prognostic factors; however, there appear to be differences in their anti-tumour functions across different solid cancer types. Notably, their tumour infiltration and cytolytic activity have been linked with a beneficial role for patient outcome in gastric cancer [121,122], hepatocellular carcinoma [68], ovarian cancer [123], colorectal cancer [124,125], renal cell carcinoma [126,127], glioblastoma [128] and triple-negative breast cancer [129,130,131]. On the other hand, they were also reported to have a negative impact on prognosis in ovarian cancer [37] and oral cancer [42], and conflicting studies also found them to positively correlate with pathogenesis in colon cancer [132] and breast cancer [133,134]. These conflicting findings appear to be mediated by the different γδ T-cell effector subtypes: anti-tumour functions generally involve γδT1 cells, and pro-tumour outcomes are linked with γδT17 cells [135]. The pro-tumour functions of γδT17 cells are discussed later in more detail.

Foord et al. demonstrated that epithelial ovarian cancer-derived γδ T cells, when stimulated, produced large amounts of IFN-γ but not IL-17 or IL-10 [123]. Furthermore, they showed that these γδ T cells effectively exercised cytolytic activities against ovarian cancer cells to a greater extent than patient-derived CD8^+^ T cells, and that patients who had more responsive γδT1 cells upon stimulation had smaller residual tumour burden and increased overall survival (OS), whereas the IFN-γ secretion capacity of αβ T cells was found to have no effect on patient survival [123]. In line with this, a recent investigation of human breast cancer patients revealed that IFN-γ-producing Vδ1^+^ TILs displayed cytotoxic capacity towards breast cancer cell lines and positively correlated with increased patient progression free survival (PFS) and OS [131]. In these patients, αβ TCR^+^ TILs were also correlated with increased PFS and, interestingly, a positive and significant correlation of tumour-infiltrating TCRα^+^ cells with Vδ1^+^ cells was reported, where the authors propose that maximal patient benefit may arise from the synergistic effect of innate-like γδ and adaptive αβ TILs [131]. In squamous cell cancer patients, Lo Presti et al. found that early stage tumours had a predominance of γδT1 TILs, and that a higher frequency of γδT1 cells correlated with a favourable patient outcome shown by an absence of relapse, lymph node invasion and mortality at follow-up [136]. Notably, in the same study, it was found that late-stage cancer patients switched to a dominance of γδT17 TILs, where a higher frequency of γδT17 cells was linked to higher relapse rates, lymph node metastasis and higher mortality rates, emphasising the functional dichotomy between these two effector subtypes in solid cancer immunity [136].

### 3.2. γδ T Cells in Haematological Malignancies

Interestingly, where the role of γδ T cells in solid tumour immunity can be conflicting, blood cancers in general appear to be more susceptible to γδ T-cell-mediated anti-tumour responses than solid tumours [137,138,139]. The cytotoxic response of γδ T cells against haematopoietic cancers in vitro has been demonstrated by numerous studies using cell lines or primary patient samples of acute myeloid leukaemia (AML) [25,140,141,142], chronic myeloid leukaemia (CML) [143], T-cell acute lymphoblastic leukaemia (T-ALL) [76,142], multiple myeloma [95,144,145], chronic lymphocytic leukaemia (CLL) [76,146], B-ALL [76,142] or other non-Hodgkin B-cell lymphomas [142,145,147]. γδ T cells were also shown to contribute to the suppression of spontaneously developing B-cell lymphomas in transgenic mice [148], and γδ T cells adoptively transferred into an Epstein–Barr virus (EVB)-induced mouse model of B-cell lymphoma significantly reduced disease burden [149].

Notably, there appear to be differences in the efficacy of haematopoietic cancer cell killing by Vδ1^+^ versus Vδ2^+^ cells. A number of studies have demonstrated anti-tumour responses of Vγ9Vδ2^+^ cells against leukaemia/lymphoma models; Vγ9Vδ2 T cells transplanted into an AML xenograft mouse model were found to localise in close proximity to engrafted leukaemic cells and significantly increase survival [141]. Vγ9Vδ2^+^ cells were also shown to have anti-tumour activity against CML cells in vitro and in vivo; however, this required the pre-treatment of CML cells with zoledronate and the administration of zoledronate plus IL-2 to mice in order to stimulate PAg expression on the tumour cells and maintain Vγ9Vδ2 T-cell activation [143]. Similarly, only around one-third of primary AML patient samples were found to be intrinsically sensitive to the anti-tumour capacity of Vγ9Vδ2^+^ cells, which was increased to 50% upon pre-treatment of the AML cells with bisphosphonates, while 50% of samples remained resistant [150]. In a clinical study administering IL-2 and pamidronate to patients with relapsed/refractory non-Hodgkin lymphoma or multiple myeloma, significant in vivo activation/proliferation of Vγ9Vδ2 T cells was difficult to achieve and partial remission was only achieved in 33% of patients, yet those that responded to the treatment demonstrated significant in vivo proliferation of Vγ9Vδ2 cells [137]. On the other hand, Vδ1 T cells have emerged as having a particular affinity for haematological cancer cell targeting. Increased circulating Vδ1^+^ T-cell numbers, associated with high IL-4 serum levels and high tumour cell expression of ULBPs, were correlated with stable disease in a 1-year follow-up in patients with low-grade non-Hodgkin B-cell lymphoma [151]. In murine xenograft models, adoptive transfer of human Vδ1^+^ T cells reduced tumour growth and dissemination in a CLL model [146], and reduced disease burden and increased survival in an AML model [25].

The anti-tumour roles of γδ T cells in haematological cancer are further exemplified in cases of allogenic haematopoietic stem cell transplantation (aHSCT) as a treatment for leukaemia/lymphoma patients. aHSCT of αβ T-cell/CD19^+^ B-cell-depleted bone marrow is now an established therapeutic protocol [152,153,154,155], where γδ T cells constitute the major T-cell population during reconstitution in the early stages post-transplantation [156]. Strikingly, a long-term follow up of acute leukaemia (ALL and AML) patients who underwent aHSCT revealed that patients who recovered with normal/low γδ T-cell levels had a 6.7 times greater risk of death, primarily from recurrent disease, than those who had increased γδ T cells [157]. The expanded γδ T-cell subtype in >90% of the surviving patients was predominately Vδ1. Higher γδ T-cell counts post-aHSCT were also correlated with improved OS in multiple myeloma [158] and AML patients [159], and were linked with a significantly reduced risk of relapse in AML patients [159]. Examination of circulating γδ T cells from acute leukaemia patients post-aHSCT revealed that cytotoxic (CD107a^+^) Vδ1^+^ cells were in higher proportions compared with their Vδ2^+^ counterparts [156].

The recognition of leukaemia/lymphoma cells by Vγ9Vδ2 T cells is expectedly induced by the recognition of PAgs (often stimulated by treatment with aminobisphosphonates) and is mediated by TCR stimulation and granule exocytosis [95,141,143,156]. Susceptibility to Vγ9Vδ2 T-cell-mediated killing has also been reported to require tumour cell expression of stress-induced molecules such as NKG2D ligand, ULBP1 [150,160], ligands for DNAM-1 [141] or the cell adhesion molecule ICAM-1 [95]. Interestingly, the picture is somewhat different for Vδ1 T-cell-mediated killing of blood cancer cells, where a strong cytotoxic response was shown to preferentially require the NKp30 receptor [76,146] and associated tumour cell expression of the NKp30 ligand, B7-H6 [25], either independently of TCR signalling [76], or involving TCR activation [25,146]. Other studies have also shown an involvement of tumour cell ULBPs or ICAM-1 expression, or activating receptors NKG2D and DNAM-1 in mediating Vδ1 cytotoxicity [144,151]. Since there is evidence that leukaemia/lymphoma cells can downregulate such ligand expression (e.g., ULBPs [151,160]), and that leukaemic stem cells do not express NKG2D ligands [161], allowing them to evade recognition by cytotoxic lymphocytes, understanding γδ T-cell target recognition and killing mechanisms will be important to establish biomarkers of potential patient sensitivity or resistance to γδ T-cell-mediated anti-tumour function. Indeed, through analysis of a panel of human blood cancer cell lines, a series of markers were previously reported to be associated with either sensitivity (ULBP1, TFR2 and IFITM1) or resistance (CLEC2D, NRP2, SELL, PKD2, KCNK12, ITGA6 and SLAMF1) to Vγ9Vδ2 T-cell-mediated cytotoxicity [142].

It is not entirely clear why haematopoietic cancer cells may be intrinsically more susceptible to γδ T-cell recognition and killing compared with solid tumours. Factors such as increased activity of the mevalonate pathway (and thereby, increased expression of PAgs), higher expression of natural cytotoxicity receptor ligands or the inherent nature of various haematopoietic cells as professional APCs to recruit T lymphocytes have been proposed [139]. It could also be influenced by the lack of recruitment or induction of ‘pro-tumour’ γδ T cells, as discussed further below. Kunzmann and colleagues revealed in a clinical study that serum levels of vascular endothelial growth factor (VEGF) were higher in patients with renal cell carcinoma and melanoma compared with AML patients, which seemed to correlate with a lack of response to Vγ9Vδ2 T-cell anti-tumour activity upon zoledronic acid administration [138]. The pro-tumour functions of IL-17-producing γδ T cells have been linked to IL-17-induced VEGF production by tumour cells to stimulate angiogenesis [162], a mechanism only relevant to solid tumours. Finally, the TME, which is vastly different between solid and haematopoietic cancers, plays a large role in shaping the functions of γδ T cells in tumour immunity.

### 3.3. γδ T Cells as Tools for Immunotherapies

The direct recognition, rapid activation and cytotoxic capacity of γδ T cells towards cancer cells makes them very attractive tools for cancer immunotherapy, as recently extensively reviewed [15,163,164]. Vδ2^+^ T cells exhibit pronounced inhibitory effects on tumourigenesis and tumour growth in a variety of malignancies and are a current hot topic in cancer immunotherapy efforts [99,165,166]. The advantage that human Vγ9Vδ2 T cells can be efficiently activated with specific cytokine and ligand stimuli (e.g., IL-2 plus aminobisphosphonates) in vitro allows for the required expansion of clinical grade autologous γδ T-cell products. However, despite the demonstrated safety of administering expanded and activated Vγ9Vδ2 T cells to cancer patients, the clinical outcomes have only shown modest therapeutic efficacy thus far [15,165]. There is also great potential for the Vδ1 subtype in cancer immunotherapy, on the one hand for their distinct ligand recognition but most importantly because they are less susceptible to T-cell exhaustion and activation-induced cell death (AICD) [167,168,169]. There have been recent advances to overcome the caveat of lacking clinical protocols to stimulate and expand Vδ1 T cells in vitro [25,146], which will now pave the way for their use in cancer immunotherapy trials [163]. Overall, γδ T-cell-based cancer immunotherapies hold high promise but still have room for improvement, and current efforts are now focusing on overcoming the main pitfalls through, for example, improving activation protocols, understanding γδ TCR diversity and receptor–ligand interactions, developing γδ TCR chimeric antigen receptor (CAR)-T cells and exploring useful drug combinations [15,163,164].

## 4. Pro-Tumour Functions of γδ T Cells

Interestingly, as mentioned above, the pro-tumour functions of γδ T cells appear to be mainly relevant in solid tumours. Tumour-promoting functions of γδ T cells are primarily mediated through the direct and indirect actions of γδT17 cells, which can be the result of polarisation programming. Indeed, the presence of high levels of IL-17 has been detected in various cancer types, such as cervical cancer [170], breast cancer [134], ovarian cancer [171], hepatocellular carcinoma [172,173], non-small cell lung cancer [174,175] and neuroblastoma [176], where it is consistently associated with tumour-promoting functions.

One mechanism by which γδ T cells have been shown to induce pro-tumour functions is by negatively regulating other cancer-killing immune cells (Figure 3). For example, Peng et al. showed that breast tumour-infiltrating Vδ1^+^ T cells supressed CD8^+^ αβ T-cell cytotoxicity and DC function [177], thus acting here as immune suppressors. Notably, co-transfer of these γδ TILs together with cytotoxic CD8^+^ αβ T cells abolished the anti-tumour effect of the CD8^+^ T cells in a melanoma mouse model [177]. Supporting these findings, Chen et al. reported that ovarian tumour-infiltrating γδ T cells recruited via the TME were primarily Vδ1^+^ γδT17 cells, which in turn secreted large quantities of pro-inflammatory IL-17 and demonstrated the capacity for immunosuppression by inhibiting CD4^+^ naïve T-cell proliferation in vitro [37].

In support of their suppressor function, murine γδ T cells stimulated in vitro with TGF-β and IL-15 were shown to express the FOXP3 transcription factor [178], resulting in immune suppressor functions similar to αβ Tregs [179]. This cytokine combination alone is not sufficient to induce human γδ Tregs from isolated peripheral blood mononuclear cell (PBMC) cultures [178]; however, additional stimulation with IPP was shown to successfully polarise Vγ9Vδ2 T cells into γδ Tregs, expressing FOXP3 and being capable of eliciting suppressive effects against stimulated PBMCs [63]. In line with this, a classical regulatory phenotype in Vδ1^+^ T cells was induced by stimulation with a plate-bound anti-Vδ1 antibody, promoting expression of regulatory markers FOXP3, CD25 and CTLA-4, as well as inducing functional suppression of CD4^+^ T-cell proliferation [180]. Additionally, TGF-β1 production by Vδ1^+^ T cells can be involved in a positive feedback loop, sustaining FOXP3 expression and also leading to production of the anti-inflammatory cytokine IL-10 [180]. Importantly, many of these cytokines are found abundantly in the TME [181], with the potential of inducing γδ T-cell polarisation, highlighting that there are multiple ways to induce a regulatory phenotype in γδ T cells, in which the TME plays a key role. This adds an important level of complexity to the roles of γδ T cells in cancer immunity that will need to be considered in γδ T-cell-based cancer immunotherapy.

γδ T cells can also regulate other immune cells to support tumour progression (Figure 3). The interaction of γδ T cells with suppressive polymorphonuclear cells (PMNs, also known as myeloid-derived suppressor cells or MDSCs) is multi-faceted; while γδT17 cells can recruit PMNs into the tumour [39], PMNs can in turn suppress cytotoxic γδT1 cells [40] and can provide a source of IL-10 and TGF-β to recruit Tregs, promoting immune suppression and tumour growth [182]. Notably, Wu et al., reported that Vδ1^+^ γδT17 cells, which were found at higher numbers in colorectal cancer, were capable of enhancing PMN migration, proliferation and survival, and were correlated with colon tumour invasiveness and progression [39]. Furthermore, neutrophils were also found to be regulated by γδT17 cells, promoting metastasis in a murine mammary tumour model [134]. Mechanistically, IL-1β from the TME was shown to elicit IL-17 production from γδ TILs, leading to the granulocyte colony-stimulating factor (G-CSF)-dependent systemic expansion of neutrophils that suppress cytotoxic CD8^+^ T cells, allowing increased metastatic disease [134].

The promotion of angiogenesis is another key tumour-promoting effect of γδT17 cells in solid cancers [43] (Figure 3). In mice, it was first demonstrated that tumour-infiltrating γδ T cells, polarised towards a γδT17 phenotype by the TME, provided the main source of tumour IL-17 necessary to induce increased levels of angiogenic factors Ang-2 and VEGF, as well as increase blood vessel numbers [162]. Pro-angiogenic functions of γδT17 cells have been reported in tumour models of ovarian cancer and human papilloma virus (HPV)-induced squamous cell carcinoma [183,184]. Notably, in gall bladder cancer patient samples, IL-17 secreted from tumour-infiltrating γδT17 cells was shown to induce tumour cell VEGF production and angiogenesis, and increased γδT17 cell numbers were associated with reduced patient survival [173].

### Regulation of γδ T-Cell Cancer Immunity

The regulation of γδ T cells can play an important role in their overall function in cancer immunity, whereby factors negatively regulating their anti-tumour activity ultimately promote tumour progression. One such example includes the hypoxia-induced immune evasion of solid cancers (Figure 3). Hypoxia can be prominent in the TME of solid tumours and was found to be a negative regulator of γδ T-cell anti-tumour function [66]. For oral cancer, Sureshbabu et al. recently demonstrated that hypoxia significantly reduced the cytolytic activity of patient-derived γδ T cells [42]. Interestingly, findings in human breast cancer revealed that γδ T cells primed by hypoxic conditions in vitro had enhanced cytolytic activities. However, this enhanced effect could not overcome the hypoxia-induced resistance of a breast cancer cell line (MCF-7) towards NKG2D-mediated γδ T-cell cytotoxicity [66]. The authors hypothesised that the increased shedding of MICA/B ligands from the tumour cell surface was causing the resistance to γδ T-cell-mediated killing. Very recently, Park et al., confirmed a similar effect of hypoxia-induced immune evasion in a model of brain cancer [64]. Again, here, they showed reduced infiltration of cytotoxic γδ T cells into tumours affected by hypoxia in the TME.

Metabolites in the TME also play an important role in regulating the infiltration and function of the different effector subtypes, γδT1 and γδT17, thereby impacting γδ T-cell tumour immune responses. γδT1 cells display a preference for glycolysis, whereas γδT17 cells are dependent on oxidative phosphorylation and have increased lipid uptake [34]. As such, it was shown that γδT1 cells pre-incubated in high glucose levels and injected into breast tumours displayed increased anti-tumour capacity. Conversely, a lipid-rich TME increased the number of γδT17 TILs, and lipid uptake promoted γδT17 cell proliferation and enhanced melanoma tumour growth [34]. Due to the high demand of cancer cells for glucose (the ‘Warburg effect’), the resulting low glucose levels in the TME, coupled with high levels of other metabolites such as lipids or lactate, which can also negatively impact T-cell effector function [185], are likely to select for specific γδ T-cell subsets in favour of reduced anti-tumour immunity and increased tumour progression [135,186]. Strategies to modulate metabolite levels in the TME would therefore be highly attractive to enhance T-cell-mediated tumour cytotoxicity.

Immune checkpoint receptors act as another important regulatory mechanism for γδ T-cell function. For instance, PD-1 is an immune checkpoint receptor expressed on most T cells, including γδ T cells, which upon ligand/antigen engagement acts as an ‘off switch’ to negatively regulate T-cell function [187]. Its ligand, PD-L1, was found to be constitutively expressed on cancer cells [188], as well as being present in the TME of several cancers [188,189,190]. As PD-1 is often present on activated and tumour-infiltrating γδ T cells, the presence of PD-L1 in the TME and expressed on tumour cells can limit γδ T-cell anti-tumour activity [188,191,192] (Figure 3). Therefore, checkpoint inhibitor immunotherapy is being explored to enhance the anti-tumour function of γδ T cells [191,193,194,195]. For example, as a strategy to boost the anti-tumour activity of Vγ9Vδ2 T cells, the use of PD-1 blockade in conjunction with PAg stimulation has been employed [196,197], aiming to neutralise the effect of the PAg stimulation-related upregulation of PD-1 on γδ T cells [198]. Interestingly, while use of anti-PD-L1 antibodies was found to indeed increase the anti-tumour effect of γδ T cells [41,194], this was not mediated by an increase in γδ T-cell cytotoxic activity, but rather by γδ T-cell-mediated ADCC induced by the tumour-targeting antibody [41]. On the other hand, PD-1 blockade was shown to significantly increase IFN-γ production in γδ T cells, although this required the prior activation of γδ T cells or sensitisation of target cells [199].

Similarly to the immune checkpoint molecules, expression of the metabolic enzyme indoleamine-2,3-dioxygenase (IDO) in cancer cells can promote an immunosuppressive shift in the TME [200]. Mechanistically, increased IDO activity promotes tryptophan catabolism, leading to a depletion of this essential amino acid in the TME and to an accumulation of tryptophan metabolites such as kynurenine, which was shown to decrease γδ T-cell cytotoxic capacity [201]. As such, IDO inhibitors were shown to improve the cytotoxicity of Vγ9Vδ2 T cells against human pancreatic [201] and breast cancer cells [202]. These findings could be particularly relevant for patients with triple-negative breast cancer, of which only a minority benefited therapeutically from PD-1 blockade [203]. Recent efforts have been made to explore the value of IDO blockade in cancer immunotherapy clinical trials [204].

Other immune cells can also negatively regulate γδ T cells (Figure 3). For example, neutrophils have been reported to suppress γδ T-cell function [205]. On the one hand, it was shown that zoledronic acid-activated neutrophils could potently inhibit peripheral blood Vγ9Vδ2^+^ T cells, which was mechanistically linked to neutrophil-derived hydrogen peroxide, serine protease and arginase-I activity [206]. Interestingly, however, a more recent study in mice found that neutrophils selectively inhibit pro-tumour γδT17 cells via production of reactive oxygen species (ROS), inducing oxidative stress and thereby mediating anti-tumour effects [207]. Therefore, the roles of neutrophils in cancer immunity appear to be pleiotropic and likely context-dependent. It was also shown in hepatocellular carcinoma patients that αβ Tregs can suppress the cytotoxic function of γδT1 cells in an IL-10- and TGF-β-dependent manner [208].

Interestingly, the local microbiota may play a role in regulating the response of γδT17 cells against tumour cells. In a genetically engineered lung cancer model driven by KRAS and p53, the lung microbiota induced the proliferation and activation of γδT17 cells to promote inflammation and tumour cell proliferation, where antibiotics or germ-free conditions suppressed these effects [175]. However, these findings are in contrast to an earlier study reporting that the lung microbiota was protective against lung tumour development by B16-F10 melanoma or Lewis lung carcinoma cells [209]. Here, it was suggested that γδT17 cells mediated this anti-tumour effect, as antibiotic treatment impaired γδT17 cell function and enhanced tumour development, and this could be reversed by adoptive transfer of untreated γδ T cells or administration of IL-17 [209].

## 5. γδ T-Cell Lymphoma/Leukaemia

The malignant transformation and outgrowth of γδ T cells seems to be a relatively rare event as most recognised T-cell neoplasms are composed of αβ T cells. Entities arising from γδ T cells include hepatosplenic T-cell lymphoma (HSTL), primary cutaneous γδ T-cell lymphoma (PCGDTL), monomorphic epitheliotropic intestinal T-cell lymphoma (MEITL), enteropathy-associated T-cell lymphoma (EATL), T-cell large granular lymphocytic leukaemia (T-LGLL) and T-ALL (Table 1). For the first three entities (HSTL, PCGDTL, MEITL), a large proportion/majority of the cases arise from γδ T cells, although αβ T-cell forms are also described, while for the latter three entities (EATL, T-LGLL, T-ALL), γδ T-cell forms are a rarity.

HSTL is an aggressive disease that primarily affects younger adults (median age of 34 years), with a higher incidence in males (≈71%) [210,211]. The disease is largely derived from γδ T cells, although up to 20% of HSTL cases express the αβ TCR [210,212], and interestingly, these cases occur more frequently in women [213]. HSTL manifests mainly in the spleen, liver and, in the majority of cases, also to a small degree in the bone marrow. The disease progresses rapidly with a median survival of 13 months [214,215]. There is a striking association with a history of therapeutic immunosuppression in HSTL.

Malignant γδ T cells can also arise in or home to the skin, and PCGDTL constitutes a separate entity among the cutaneous T-cell lymphomas (CTCLs) [212]. PCGDTL was formerly a subset of subcutaneous panniculitis-like T-cell lymphoma (SPTCL), which is now only reserved for the αβ cases, while the PCGDTLs are prognostically distinct [216]. This rare disease makes up < 1% of all primary CTCLs and has a highly aggressive course, with a median survival of 15–31 months [217,218]. γδ T-cell-derived CTCLs have a significantly lower median survival as compared to αβ TCR^+^ CTCL patients (15 months and 166 months, respectively) [218]. PCGDTL is also associated with a slight male predominance [217,218].

The intestinal T-cell lymphomas EATL and MEITL are also rare but aggressive diseases [212]. EATL is strongly associated with celiac disease and is predominantly of the αβ T-cell subtype [219,220], although cases of γδ TCR^+^ EATL have been reported [221,222]. MEITL, on the other hand, has no association with celiac disease, and a larger proportion of cases can arise from γδ T cells, varying in reports ranging from 25–80% of cases expressing γδ TCR, with the remaining cases having an αβ phenotype, presenting as TCR silent or, interestingly, co-expressing both αβ and γδ TCRs [223,224,225]. Both EATL and MEITL present more frequently in males (EATL with a slight male predominance [226,227,228], MEITL with a male to female ratio of 2:1 [223,224,225,229]). As aggressive and treatment-refractory diseases, both EATL and MEITL show a median OS of 7 months [223,224,226,230].

LGLL is a rare lymphoproliferative disorder that can arise from T or NK cell lineages, and unlike the aforementioned diseases, it has a rather indolent course. Nevertheless, the associated cytopenias and autoimmune phenomena make it a highly symptomatic disease. The most frequent form of T-LGLL (≈85% of cases) is derived from αβ TCR^+^ CD8^+^ cytotoxic T cells, where chronic NK-LGLL comprises <10% of cases [231]. γδ T-LGLL represents an even smaller subset of the overall cases, and, perhaps surprisingly given the aggressive nature of most other γδ T-cell derived malignancies, γδ T-LGLL has a clinical presentation and an indolent course that is similar to its αβ TCR^+^ counterpart (62–114 months median OS for both) [232,233,234]. No sex-specific biases in T-LGLL diagnoses have been found [232,233,235].

T-ALL arises from the malignant transformation of immature T cells in the thymus or bone marrow, and it occurs most frequently in children and young adults. Approximately 10–15% of all T-ALL cases express the γδ TCR [236], which is perhaps unexpectedly high given that γδ thymocytes comprise only ≈1% of T cells in the human thymus [237], suggesting that γδ thymocytes may be more susceptible to transformation compared to αβ thymocytes [16]. Notably, the 5-year OS for patients with γδ T-ALL was significantly lower as compared to other T-ALL patients (66.7% vs. 95.7%) [237], and increased splenomegaly and white blood cell counts were reported in adult γδ T-ALL patients [236]. Reports also suggest that there is an association of male predominance in γδ T-ALL [236,237].

### 5.1. Development of γδ T-Cell Lymphoma/Leukaemia

The transformation process of a γδ T cell into a malignant clone is, like for other cell types, a complex and multi-factorial process. In contrast to many other types of haematopoietic cells, T-cell oncogenesis involves overcoming the safeguarding mechanism of TCR/MHC niche-mediated cross-clonal control, which is a part of T-cell homeostasis [238]. On the basis of disease aetiology across all γδ T-cell lymphoma/leukaemia subtypes, there is evidence that a chronic inflammatory or immune-suppressed environment and persistent activation signals are key triggers for γδ T-cell transformation.

Around 20% of HSTL cases arise in patients with pre-existing immune dysregulatory disorders, such as inflammatory bowel disease (IBD), autoimmune disorders (systemic lupus, rheumatoid arthritis), infections (malaria), haematological malignancies (Hodgkin lymphoma) or long-term immunosuppression after organ transplantation [210,211,239]. Immunosuppressive therapies given to patients with such conditions, such as thiopurines (e.g., azathioprine or 6-mercaptopurine [6-MP]) for IBD patients, have been linked to the development of HSTL [240,241]. Concurrent administration of other agents such as TNF-α inhibitor therapy appears to increase the risk of HSTL development, although this observation may be confounded by the need for increased therapy in patients with more severe conditions, increased inflammation and chronic antigenic stimulation [210,240]. EATL is strongly linked with celiac disease, and therefore with autoimmunity and inflammation of the intestine, although the majority of cases derive from αβ T cells [219,220]. A large study of 53 PCGDTL patients also revealed that additional pre-existing conditions involved in immune dysregulation were common, such as lymphoproliferative disorders (Hodgkin lymphoma, B-cell non-Hodgkin lymphoma, CLL), hypothyroidism secondary to Hashimoto thyroiditis, Crohn’s disease, alopecia areata, celiac disease/uveitis/arthritis and sarcoidosis [217]. Fitting the concept of narrowed clonal repertoires promoting the outgrowth of malignant T-cell clones (homeostasis concept) [238], any therapy that reduces the spectrum of clonal T-cell diversity may promote γδ T-cell oncogenesis. T-LGLL is also strongly linked to co-existing autoimmune diseases, with the most common being rheumatoid arthritis, and others including systemic lupus erythematosus, Felty’s syndrome, chronic IBD, autoimmune haemolytic anaemia and other haematological neoplasms (myelodysplastic syndrome, B-cell malignancies, aplastic anaemia) [231,242]. Studies comparing αβ and γδ T-LGLL revealed that the occurrence of such immune dysregulatory conditions is similar in patients with either subtype [232,233,234].

Therefore, at least in mature γδ T cells, the co-occurrence of inflammatory or immune-modulating disorders in a subset of patients with γδ T-cell lymphoma/leukaemia, irrespective of their origin or main site of infiltration, suggests that chronic inflammation and activation signals could be common states driving the transformation process. Of course, these diseases also arise de novo in many cases, but such inflammatory or immunosuppressive environments may accelerate initiating events (e.g., through accumulating reactive oxygen species) to allow damaged cells to persist. Indeed, we previously reported that patients with newly diagnosed rheumatoid arthritis have expanded CD8^+^ T-cell clones harbouring somatic mutations linked to cell proliferation [243], and also that STAT3 mutations were significantly associated with rheumatoid arthritis in T-LGLL patients [244]. Notably, particularly IL-17-producing γδ T cells have been implicated in the pathogenesis of inflammatory and autoimmune diseases [245], highlighting the possibility of a self-promoting environment ultimately favouring transformation.

The specific γδ T-cell subsets that undergo transformation to drive these diseases vary across the different lymphoma/leukaemia entities. In HSTL patients, 90% of cases are derived from the Vδ1 subset [210], which is the main γδ T-cell subtype occupying both the liver and spleen in humans (Figure 1). The same is true for γδ T-ALL, in which 80% of patients have a Vδ1^+^ disease [246]. This is marginally higher but closely in line with what would be expected on the basis of normal healthy proportions in the human thymus, where 60–75% of γδ T cells are Vδ1^+^ and 20–25% are Vδ2^+^ [247,248]. A report on a small number of patients revealed that patients with MEITL also present with Vδ1^+^ disease [249], which is perhaps expected given that the Vδ1 subtype is also the most prevalent in mucosal tissues such as the intestine (Figure 1). Interestingly, while it was previously thought that PCGDTLs arise from the Vδ2 subtype [250], recently, Daniels et al. interrogated different layers of the skin to demonstrate that disease arising from the upper epidermal/dermal layers is specifically of the Vδ1 subtype, whereas disease originating from lower subcutaneous adipose tissue is Vδ2^+^, consistent with their normal distribution in healthy skin [251]. The authors also revealed that the Vδ1 lymphomas had an accompanying Vγ3 or Vγ5 chain, whereas unexpectedly, all Vδ2 cases had an accompanying Vγ3 chain [251]. No differences in patient prognosis between the two subtypes were identified [251]. In cases of γδ T-LGLL, both subtypes have been reported, with a distribution of 48–70% Vδ2^+^ and 20–43% Vδ1^+^ [232,233]. This is again mostly in line with the proportions of γδ T-cell subsets in healthy adult peripheral blood, which are reported as 80–85% Vδ2^+^ and 10–20% Vδ1^+^ [232], with a potential slight bias towards Vδ1 cells in γδ T-LGLL. Interestingly, there have been some reports of rare γδ T-cell lymphoma/leukaemia cases arising from Vδ3 or Vδ6 subsets in patients with MEITL, γδ T-LGLL and γδ T-ALL [232,246,252]. Overall, it appears that the γδ T cell of origin in these diseases is largely dictated by the physiological proportion of γδ T-cell subsets in the tissue where the disease arises.

### 5.2. Common Genetic Aberrations

Irrespective of the sites/tissues in which γδ T-cell transformation takes place, the genetic aberrations that frequently occur across the various γδ T-cell lymphoma/leukaemia entities have distinct commonalities. Interestingly, gains of chromosome 7q are observed in patients from almost all γδ T-cell cancer subtypes (Table 1). Notably, isochromosome 7q (gain of 7q and concomitant loss of 7p) is detected in up to 70% of HSTL patients [210,240,253], highlighting that this event is an early driver of disease development. It is not completely understood how 7q gains drive the pathogenesis of HSTL, but it has been linked to the upregulation of genes including *ABCB1*, *RUNDC3B* and *PPP1R9A* [254]. Various other chromosomal losses are also observed across γδ T-cell lymphoma/leukaemia patients (Table 1), with losses of 9p occurring in PCGDTL, EATL and MEITL patients, potentially linked to the loss of tumour suppressors *CDKN2A/B* [255]. In line with αβ T-ALL and other acute leukaemias, complex cytogenetic abnormalities/translocations are also common in γδ T-ALL (Table 1). Interestingly, the fusion-proteins SET-NUP214 and CALM-AF10 were identified as specific to γδ T-ALL and were found to be associated with chemotherapy resistance and poor prognosis, respectively [256,257]. Chromosomal aberrations (gains, losses, translocations) may therefore represent common, primary initiating events in γδ T-cell transformation, predisposing the γδ T cells to acquiring additional driving mutations. This phenomenon is also observed in other mature T-cell malignancies, such as recurrent rearrangements at chromosome 14 in T-cell prolymphocytic leukaemia (T-PLL) and the driver fusion-protein oncogene NPM-ALK in anaplastic large cell lymphoma (ALCL) [258,259].

Strikingly, there are clear commonalities in the signalling pathways most frequently and recurrently affected by somatic mutations across the γδ T-cell malignancies, with the key pathways affected including JAK-STAT, epigenetic regulation, RAS-MAPK and AKT-mTOR (Table 1). Perhaps unsurprisingly, these pathways are all critical for γδ T-cell development, effector function and/or TCR signalling [33,260,261,262]. The hyperactivation of JAK-STAT, RAS-MAPK and AKT-mTOR signalling would therefore confer persistent proliferative and survival signals to the γδ T cancer cells, whereas alterations to chromatin modifier proteins, frequently seen also in other haematopoietic cancers [263], can further dysregulate the expression of oncogenes and tumour suppressors and/or alter lineage-specific factors via manipulation of the chromatin landscape. The latter, particularly altered genomic methylation profiles, appears to be an especially important driver in γδ T-cell cancers, as evidenced by the relatively frequent loss-of-function mutations observed in the histone methyltransferase *SETD2* and the DNA demethylase *TET2* (Table 1). In addition, there is a clear dominance of hyperactivating *JAK-STAT* gene mutations across the γδ T-cell lymphoma/leukaemia entities, particularly in *STAT5B*, *STAT3, JAK3* and *JAK1.* Notably, STAT5 and STAT3 also have clear roles in regulating the epigenetic and chromatin landscape [264]. Indeed, we have shown that the most frequent gain-of-function *STAT5B* mutation in T-cell cancers, N642H [265], induces considerable alterations to DNA methylation as well as transcriptional profiles in T cells [266], and is sufficient to drive the transformation of γδ T cells and induce γδ T-cell neoplasia in mice [267].

**Table 1 cancers-13-06212-t001:** Cancer entities arising from γδ T cells and their key genetic aberrations.

Disease Subtype	Median Survival	Disease Site	Chromosomal Lesions	Dysregulated Pathways	Genes Frequently Affected	Ref.
**HSTL**	13 months	Spleen, liver	Isochromosome 7q, trisomy 8	Epigenetic modifiers	*SETD2, INO80, TET3, SMARCA2*	[214] [215] [268]
				JAK-STAT	*STAT5B, STAT3*	
				AKT-mTOR	*PIK3CD*	
**MEITL**	7 months	Intestine	Gain of 8q24 (*MYC*), 1q, 7q or 9q; loss of 8p, 16q, 11q or 9p21.3 (*CDKN2A/B*) *	JAK-STAT	*STAT5B, JAK1, JAK3 **	[222] [229] [269] [255]
				RAS-MAPK	*KRAS, NRAS, BRAF **	
				Epigenetic modifiers	*SETD2, TET2, YLPM1, CREBBP **	
**PCGDTL**	15–31 months	Skin	Gain of 1q, 15q or 7q; loss of 9p or 18q	RAS-MAPK	*KRAS, NRAS, MAPK1*	[217] [218] [251]
				JAK-STAT	*STAT5B, SOCS1, JAK3, STAT3*	
				Epigenetic modifiers	*ARID1A,* *TRRAP, TET2, KMT2D*	
				Cell cycle	*CDKN2A*	
**γδ** **T-LGLL**	62–114 months	Blood	Rare; gain of 7p21 and loss of Chr Y or Chr 6 reported	JAK-STAT	*STAT3, STAT5B **	[233] [232] [234] [244]
**EATL**	7 months	Intestine	Gain of 1q, 7q or 9q; loss of 9p or 17p12- 13.2 (*TP53*) *	JAK-STAT	*SOCS1, JAK1, STAT3, JAK3, STAT5B **	[222] [219] [269]
				Epigenetic modifiers	*TET2, SETD2 **	
				Survival	*DAPK3 **	
**γδ** **T-ALL**	5-year OS: 67%	Thymus, blood	Complex cytogenetic abnormalities; loss of 6q13- 23 or 12p11-13; translocations t(11;14) or t(10;11)	-	Gene fusions: *SET-NUP214, CALM-AF10*	[237] [256] [257] [270] [271]
				JAK-STAT	*IL7R, JAK3, STAT5B, JAK1 **	
				AKT-mTOR (via CK2)	-	

* including, but not restricted to, γδ TCR^+^ cases due to their rarity in the disease and/or lack of specific analyses. OS, overall survival; Chr, chromosome; CK2, casein kinase 2; HSTL, hepatosplenic T-cell lymphoma; MEITL, monomorphic epitheliotropic intestinal T-cell lymphoma; EATL, enteropathy-associated T-cell lymphoma; PCGDTL, primary cutaneous γδ T-cell lymphoma; T-LGLL, T-cell large granular lymphocytic leukaemia; T-ALL, T-cell acute lymphoblastic leukaemia.

### 5.3. Current Treatment Options and Promising Therapeutic Directions

Despite T-cell neoplasms being a heterogeneous disease group, many of our current chemotherapy approaches are rather uniformly adopted from those developed for aggressive B-cell lymphomas [272,273,274,275,276]. We have just recently begun to address certain entities more specifically. Generally, current standard regimens of chemotherapy, including those applied to γδ T-cell lymphoma/leukaemia, are mostly anthracycline-based and can additionally involve substances like etoposide, ifosfamide, methotrexate, asparaginase and others [274,277,278,279,280,281,282,283]. The therapeutic goal at first-line treatment is to achieve a complete remission (CR) status, which can then be followed by a consolidating autologous or allogeneic stem cell transplantation in eligible patients. This to date remains the only effective, potentially curative treatment modality. Such strategies of early intensified therapy are particularly applied to the high-risk diseases EATL, MEITL, HSTL and PCGDTL. In some cases, successful long-term remission following this strategy has been reported for γδ HSTL patients [284,285]. Unfortunately, however, a considerable number of patients do not respond well to the current chemotherapy treatments available, and thus fail to qualify for these consolidating measures [211,253,286]. Clinical investigations of new therapies for γδ T-cell lymphoma patients are challenging due to the rarity of the diseases, imposing difficulties in patient recruitment and tumour sampling for translational studies. Furthermore, to date, there are no robust pre-clinical models for γδ T-cell lymphoma available to facilitate testing of targeted therapies. 

Due to the lack of targeted therapy options, treatments approved for other (mainly αβ) peripheral T-cell lymphomas (PTCLs) or B-cell lymphomas are repurposed for use in γδ T-cell lymphoma/leukaemia patients. Newer, more targeted treatment options that are clinically approved are being explored in trials for relapsed/refractory PTCL or as combinations in first-line settings. These trials include patients with γδ T-cell lymphoma/leukaemia, as trials dedicated only to these entities are usually not feasible. The substance classes being tested include folate metabolite inhibitors (methotrexate, pralatrexate), oral inhibitors of phosphoinositide-3 kinase (PI3K; duvelisib, copanlisib, tenalisib) [287], JAK inhibitors, antibody-drug conjugates (brentuximab vedotin; only FDA-approved when CD30 is expressed in >10% of tumour cells), histone deacetylase (HDAC) inhibitors (romidepsin, belinostat, chidamide) [288] or DNA-demethylating agents. As a notable example for the efficacy of HDAC inhibitors, Wang et al. recently reported an HSTL case where conventional chemotherapy as induction therapy failed to control disease progression [289]. Strikingly, upon treatment with oral chidamide combined with chemotherapy (ifosfamide, carboplatin, etoposide), followed by chidamide maintenance, the patient achieved a CR for 9 months. Other much older alternative treatment options for unresponsive cases of HSTL include purine analogues. There have been several reports of successful treatment with 2’-deoxycoformycin (pentostatin) that resulted in short-term relief from symptoms and tumour cell clearance in blood [290,291,292,293]. There are newer purine analogues such as cladribine that present with a better cytotoxic profile and that are also being explored for further use in some other PTCLs.

Overall, new therapeutic options are urgently needed for these aggressive γδ T-cell diseases. Given the more recent advances in understanding the molecular profiles and events driving these malignancies (Table 1) [215,251,269,294], it is perhaps not surprising that HDAC and PI3K inhibitors show efficacy, and drugs targeting the JAK-STAT or RAS-MAPK pathways also hold promise in certain γδ T-cell cancer entities. The more we understand the driving mechanisms and molecular properties of these rare diseases, the better we can move forward in designing specifically targeted therapies to maximise patient outcomes. However, it must be kept in mind that the greatest challenge for clinical trials in these rare entities remains the limited patient numbers and resulting difficulties to recruit enough patients at the study site level. Therefore, advancements in suitable pre-clinical models as well as the compassionate use of potentially suitable targeted therapies will be important, and efforts should be made to make drug development more attractive for these rare diseases.

## 6. Conclusions

Recent years have seen a large step forward in both our understanding of the roles of human γδ T cells in cancer immunity, as well as in our understanding of the genetic and molecular basis of aggressive γδ T-cell leukaemia/lymphoma. These new insights will pave the way for considerable advancements in the years ahead, especially regarding the development of Vδ1 T cells as tools for adoptive γδ T-cell immunotherapy for cancer patients. Nevertheless, it will be important to continue to dissect the pro-tumour functions of γδT17 cells, particularly regarding polarisation triggers in the human context and the negative impact this could play on their therapeutic utility. Furthermore, new therapeutic strategies are urgently needed for patients suffering from incurable γδ T-cell leukaemia/lymphoma, and current research focusing on identifying and targeting common dysregulated pathways, as well as the development of new faithful pre-clinical models, should hopefully soon lead to new breakthroughs to help these patients.

## Figures and Tables

**Figure 2 cancers-13-06212-f002:**
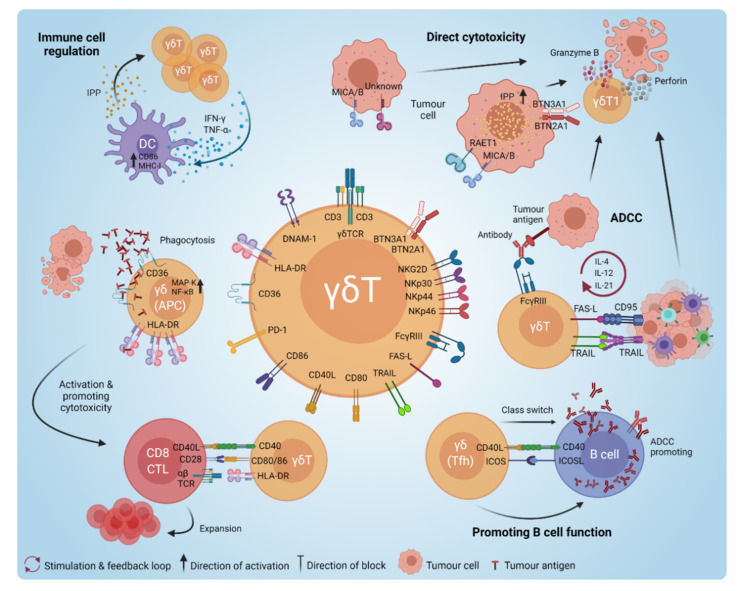
Anti-tumour functions of γδ T cells. Direct cytotoxicity is induced by detection of intracellular stress signalling in tumour cells (i.e., RAET1, MICA/B ligands, IPP production), engagement of the γδ TCR and death receptors (i.e., TRAIL, FAS) and granzyme B/perforin release. Antibody-dependent cellular cytotoxicity (ADCC) via the FcγRIIIA receptor can also induce target cell death. γδ T cells can promote anti-tumour functions indirectly through immune cell regulation, via promoting B-cell function and class switch to antibody production; interaction/promotion of dendritic cell (DC) activity; acting as an antigen presenting cell (APC) via scavenger receptor activity (i.e., CD36) and promoting CD8^+^ cytotoxic T lymphocyte (CTL) function via CD40, CD80 and HLA-DR engagement. IPP, isopentenyl pyrophosphate.

**Figure 3 cancers-13-06212-f003:**
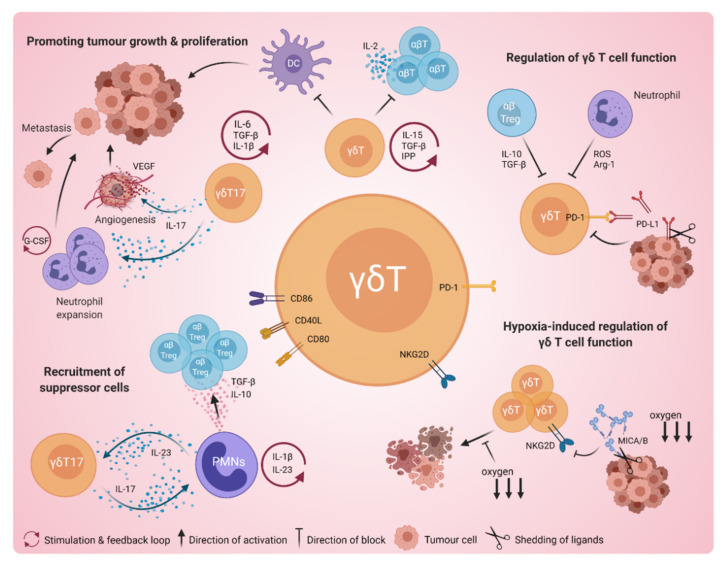
Pro-tumour functions of γδ T cells and the negative regulation of their anti-tumour capacity. Pro-tumour functions of γδ T cells include direct immune suppressor functions blocking αβ T-cell cytotoxicity and DC maturation, promoting angiogenesis and stimulating immune-suppressive neutrophil expansion. γδ T cells can recruit/promote suppressive polymorphonuclear cells (PMNs), facilitated by IL-17, IL-23 and IL-1β feedback loops. Negative regulation of γδ T-cell anti-tumour activity can occur via PD-1/PD-L1 interaction induced by tumour cells, as well as suppressive activity by neutrophils and αβ Tregs. Hypoxia-induced regulatory effects from the tumour microenvironment (TME) can induce tumour cell shedding of MICA/B to block NKG2D-mediated γδ T-cell cytotoxicity. Arg-1, arginase-I; G-CSF, granulocyte colony-stimulating factor; ROS, reactive oxygen species; TGF-β, transforming growth factor-β; VEGF, vascular endothelial growth factor.

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
