# Peer review of "The Diverse Roles of γδ T Cells in Cancer: From Rapid Immunity to Aggressive Lymphoma"

_cancers, 2021, doi:10.3390/cancers13246212_

Round 1
Reviewer 1 Report
Very extensive and comprehensive review, about gamma delta T cell in cancer. The review summarizes very well the pro and anti tumoral effect of the gamma delta T cell. The paper is based on 288 references including key publication in high impact factor journal. It is a clear update on all the current knowledge on this specific T cell subtype.
Author Response
Comment: Very extensive and comprehensive review, about gamma delta T cell in cancer. The review summarizes very well the pro and anti tumoral effect of the gamma delta T cell. The paper is based on 288 references including key publication in high impact factor journal. It is a clear update on all the current knowledge on this specific T cell subtype.
Response: We sincerely thank the reviewer for their time and positive assessment of our manuscript.
Reviewer 2 Report
This review article is well written about classification and anti- or pro-tumor function of γδT-cells in detail and also widely explain γδT-cell transformation, T-cell leukaemia/lymphoma entities, their genetic and molecular profiles as well as current and future treatment strategies with 288 of references. I believe that this review is helpful for clinicians and researchers to understand current status of biology and oncogenesis of T-cells, and development of novel immunotherapy based on γδT-cells.
Author Response
Comment: This review article is well written about classification and anti- or pro-tumor function of γδT-cells in detail and also widely explain γδT-cell transformation, T-cell leukaemia/lymphoma entities, their genetic and molecular profiles as well as current and future treatment strategies with 288 of references. I believe that this review is helpful for clinicians and researchers to understand current status of biology and oncogenesis of T-cells, and development of novel immunotherapy based on γδT-cells.
Response: We sincerely thank the reviewer for their time and positive assessment of our manuscript.
Reviewer 3 Report
The authors present an interesting and comprehensive review of gd T cells in cancer. The manuscript is well written and informative. I would suggest only some minor changes.
Section 2 is titled “gd T cells as effectors in cancer immunity”, although most part of the section deals with the functional subtypes and regulation of gd T cells, not necessarily associated with cancer. It would make more sense to have one section with the physiology of gd T cells, and then one section dedicated predominantly to the function in cancer.
Is there a connection between the gd T cell subtypes related to the receptor type (Vd1 vs Vd2) and the functional subtypes (gdT1 vs gdT17)? This should be clarified.
In section 2, the TME section could be expanded with more information. The authors mention the role of VEGF and hypoxia. It would be nice to have more information on the effect of nutrient deprivation, for instance the effect of glucose deprivation on the metabolic switch between gdT1 and gdT17 phenotypes, as well as the effect of other metabolites typically present in the tumor microenvironmentm such as lactic acid, which has a known immunomodulation function.
In line 212, the mechanism of gd T cell stimulation by IPP is not very clear. Is isopentenyl pyrophosphate released from cancer cells and stimulate gd T cells? If so, how is IPP released from cancer cells? Is it actively transported, or is it released as a consequence of cell death? Or is it translocated to the outside layer of the plasma membrane of cancer cells? The authors mention it is also a microbial product. Is there a role of the lung microbiome in the stimulation of gd T cells against lung cancer?
At lines 276-279, the authors state that the pro- or anti-tumor function of gd T cells may be associated with the subtypes gdT17 and gdT1, respectively. However, there are no references cited to support this observation.
At line 384 the authors state that the pro-tumor functions of IL-17-producing gd T cells have been linked to induction of VEGF produciton by tumor cells, suggesting that VEGF stimulates in some way the gdT17 phenotype. At line 472 the authors state that gd T cells stimulate angiogenesis because IL-17 stimulates VEGF. The mutual relationship between gd T cells and VEGF shold be better explained and clarified. Does VEGF stimulate gd T cells, or do gd T cell enhance the production of VEGF? Or are both mechanisms active at the same time?
At line 515, IDO is strictly not an immune checkpoint molecule, but a metabolic enzyme involved in triptophan metabolism. The mechanism of action of IDO inhibitors shold be explained better.
Author Response
Comment: The authors present an interesting and comprehensive review of gd T cells in cancer. The manuscript is well written and informative. I would suggest only some minor changes.
Response: We sincerely thank the reviewer for their time and positive assessment of our manuscript.
Comment: Section 2 is titled “gd T cells as effectors in cancer immunity”, although most part of the section deals with the functional subtypes and regulation of gd T cells, not necessarily associated with cancer. It would make more sense to have one section with the physiology of gd T cells, and then one section dedicated predominantly to the function in cancer.
Response: We agree and thank the reviewer for this suggestion. We have now changed the title of Section 2 (line 113) to “Functional γδ T-cell subsets in mice and humans” to better reflect the topics discussed in this section.
Comment: Is there a connection between the gd T cell subtypes related to the receptor type (Vd1 vs Vd2) and the functional subtypes (gdT1 vs gdT17)? This should be clarified.
Response: There is no clear association between the Vδ chain expressed and the functional subtype of human γδ T cells. Vδ1, Vδ2 and Vδ3 T cells have all been shown to have the capacity for IL-17 production, but they are all much more frequently found as cytotoxic IFN-γ-producing cells. In mice, there is a clearer association between specific Vγ chain expression and functional subtypes, e.g. Vγ4 and Vγ6 subsets are biased towards IL-17 production whereas Vγ1, Vγ5 and Vγ7 subsets are generally IFN-γ-producing. We discuss this topic regarding human γδ T cells on lines 150-158, and to improve the clarity we have now added the statement: “In contrast to murine γδ T cells, there is no association between TCR Vδ chain usage and the functional subtype of human γδ T cells”. Additionally, for completeness, we have also now included a statement pointing out the clear association in murine cells (lines 125-127).
Comment: In section 2, the TME section could be expanded with more information. The authors mention the role of VEGF and hypoxia. It would be nice to have more information on the effect of nutrient deprivation, for instance the effect of glucose deprivation on the metabolic switch between gdT1 and gdT17 phenotypes, as well as the effect of other metabolites typically present in the tumor microenvironmentm such as lactic acid, which has a known immunomodulation function.
Response: We thank the reviewer for this suggestion to expand the TME section and include information on metabolism. We have now added the following paragraph to the manuscript (lines 508-521):
“Metabolites in the TME also play an important role in regulating the infiltration and function of the different effector subtypes, γδT1 and γδT17, thereby impacting γδ T-cell tumour immune responses. γδT1 cells display a preference for glycolysis whereas γδT17 cells are dependent on oxidative phosphorylation and have increased lipid uptake [34]. As such, it was shown that γδT1 cells pre-incubated in high glucose levels and injected into breast tumours displayed increased anti-tumour capacity. Conversely, a lipid-rich TME increased the number of γδT17 TILs, and lipid uptake promoted γδT17 cell proliferation and enhanced melanoma tumour growth [34]. Due to the high demand of cancer cells for glucose (the “Warburg effect”), the resulting low glucose levels in the TME coupled with high levels of other metabolites like lipids or lactate, which can also negatively impact T-cell effector function [185], are likely to select for specific γδ T-cell subsets in favour of reduced anti-tumour immunity and increased tumour progression [135,186]. Strategies to modulate metabolite levels in the TME would therefore be highly attractive to enhance T-cell-mediated tumour cytotoxicity.”
Comment: In line 212, the mechanism of gd T cell stimulation by IPP is not very clear. Is isopentenyl pyrophosphate released from cancer cells and stimulate gd T cells? If so, how is IPP released from cancer cells? Is it actively transported, or is it released as a consequence of cell death? Or is it translocated to the outside layer of the plasma membrane of cancer cells? The authors mention it is also a microbial product. Is there a role of the lung microbiome in the stimulation of gd T cells against lung cancer?
Response: We thank the reviewer for bringing this to our attention. The current accepted model for IPP-mediated activation of γδ T cells is such that intracellularly accumulated IPP binds to the intracellular domain of the butyrophilin molecule BTN3A1, causing a conformational change and interaction with BTN2A1, allowing binding of this complex by the T cell receptor of Vγ9Vδ2 cells. There are also reports that IPP can be released from cancer cells to activate Vγ9Vδ2 cells, and the mechanism of IPP release was shown to involve the ATP-binding cassette transporter A1 (ABCA1) in cooperation with apolipoprotein A-I (apoA-I) and BTN3A1. We have now included these extra details and clarification to the relevant section (lines 221-228). We also noticed upon revising that our discussion on MICA/B and NKG2D in this section on IPP was perhaps misleading, and we have now moved this information earlier in the paragraph for better clarity (line 213-216).
Regarding a role of the lung microbiome in the stimulation of γδ T cells against lung cancer, yes there have been studies examining this with somewhat conflicting results. In a genetically engineered lung cancer model driven by KRAS and p53, the lung microbiota induced the proliferation and activation of γδT17 cells to promote inflammation and tumour cell proliferation, where antibiotics or germ-free conditions suppressed these effects. However, these findings are in contrast to an earlier study reporting that the lung microbiota was protective against lung tumour development by B16/F10 melanoma or Lewis lung carcinoma cells. Here, it was suggested that γδT17 cells mediated this anti-tumour effect, as antibiotic treatment impaired γδT17 cell function and enhanced tumour development, and this could be reversed by adoptive transfer of untreated γδ T cells or administration of IL-17. We have now included this information in the manuscript (lines 562-571).
Comment: At lines 276-279, the authors state that the pro- or anti-tumor function of gd T cells may be associated with the subtypes gdT17 and gdT1, respectively. However, there are no references cited to support this observation.
Response: We have now included a reference to this sentence (line 288) to support this statement. Additionally, we feel that this is a major theme in our review and that all the literature we summarize in both the anti- and pro-tumor sections further support this statement.
Comment: At line 384 the authors state that the pro-tumor functions of IL-17-producing gd T cells have been linked to induction of VEGF produciton by tumor cells, suggesting that VEGF stimulates in some way the gdT17 phenotype. At line 472 the authors state that gd T cells stimulate angiogenesis because IL-17 stimulates VEGF. The mutual relationship between gd T cells and VEGF shold be better explained and clarified. Does VEGF stimulate gd T cells, or do gd T cell enhance the production of VEGF? Or are both mechanisms active at the same time?
Response: Indeed, the secretion of IL-17 by gdT17 cells has been shown to induce the production of VEGF by cancer or stromal cells, promoting angiogenesis. This mechanism has been reported in multiple studies, which we describe in detail in lines 482-491. We are not aware of any evidence that VEGF acts upon gdT17 cells to regulate their pro-tumor function. To bring more clarity to this pro-angiogenic mechanism of IL-17, we have re-worded the sentence on line 395 (previously 384), as well as parts on lines 485-490, and we have amended Figure 3 to make the IL-17-mediated mechanism clearer.
Comment: At line 515, IDO is strictly not an immune checkpoint molecule, but a metabolic enzyme involved in triptophan metabolism. The mechanism of action of IDO inhibitors shold be explained better.
Response: We thank the reviewer for pointing out this error. We have now revised our description of IDO and have included additional information on the mechanism of action of IDO and IDO inhibition on γδ T cell anti-cancer immunity (lines 540-550):
“Similarly to the immune checkpoint molecules, expression of the metabolic enzyme indoleamine-2,3-dioxygenase (IDO) in cancer cells can promote an immunosuppressive shift in the TME [200]. Mechanistically, increased IDO activity promotes tryptophan ca-tabolism leading to a depletion of this essential amino acid in the TME and to an accumu-lation of tryptophan metabolites such as kynurenine, which was shown to decrease γδ T-cell cytotoxic capacity [201]. As such, IDO inhibitors were shown to improve the cytotoxi-city of Vγ9Vδ2 T cells against human pancreatic [201] and breast cancer cells [202]. These findings could be particularly relevant for patients with triple negative breast cancer, of which only a minority benefited therapeutically from PD-1 blockade [203]. Recent efforts have been made to explore the value of IDO blockade in cancer immunotherapy clinical trials [204].”